# Healthcare Professionals’ Perceptions and Concerns towards Domestic Violence during Pregnancy in Southern Italy

**DOI:** 10.3390/ijerph16173087

**Published:** 2019-08-25

**Authors:** Fortuna Procentese, Immacolata Di Napoli, Filomena Tuccillo, Alessandra Chiurazzi, Caterina Arcidiacono

**Affiliations:** Department of Humanities, University of Naples, 80133 Naples, Italy

**Keywords:** domestic violence, health professionals, pregnancy, health department procedures

## Abstract

Background: Literature on pregnancy highlighted a large number of women abused by their partners, especially among low-income teenagers attending hospital for pregnancy check-ups. Pregnancy represents a key moment for diagnosing domestic violence. Method: This study explores health professionals’ perceptions and concerns about domestic violence against women in services dealing with pregnant women. The twenty-four interviewees were from an Obstetrical-Gynecological walk-in Clinic in the south of Italy. The textual data has been complementarily analyzed by means of two different procedures: Symbolic-structural semiotic analysis and Thematic content analysis. Results: What emerges is that the interviewees of the clinic do not regard the issue of domestic violence as a matter of direct interest for the health service. The clinic is seen as a place for urgent contact, but one where there is not enough time to dedicate to this kind of patient, nor an adequate space to care for and listen to them. Obstetricians and health personnel expressed a negative attitude when it comes to including questions regarding violence and abuse in pre-natal reports. Training for health and social professionals and the empowering of institutional support and networking practices are needed to increase awareness of the phenomenon among the gynecological personnel.

## 1. Introduction

This article deals with domestic violence and intimate partner violence (IPV) during pregnancy. There is, in fact, an increase in evidence of the violence perpetrated against women during this period [1]. Usually the risk of being subjected to IPV increases among young girls, women belonging to an ethnic minority, the poorly educated, those of low socioeconomic status, the unemployed, and those presenting limitations in everyday activities [1,2,3,4,5,6,7,8]. Pregnancy constitutes one of those periods in a woman’s life-cycle in which the risk of being a victim of domestic violence increases considerably.

According to a WHO (World Health Organization) report “studies have found substantial levels of physical IPV during pregnancy in settings around the world. The WHO multi-country study found prevalence of physical IPV in pregnancy ranging from 1% in urban Japan to 28% in provincial Peru, with prevalence in most sites of 4–12%. Similarly, a review of studies from 19 countries found prevalence ranging from 2% in settings such as Australia, Denmark and Cambodia, to 13.5% in Uganda, with the majority ranging between 4% and 9%. A few facility-based studies in some settings have found even higher prevalence s, including one from Egypt with an estimated prevalence of 32% and a review of studies from Africa that found a prevalence as high as 40% in some settings” ([9], p.6). While only a few studies have followed pregnant women prospectively to examine the impact of violence on birth outcome, and one of them, a prospective study of pregnancy, highlighted a large number of women abused by their partners, especially among low-income teenagers attending hospital for pregnancy check-ups [10].

A female victim of violence during pregnancy suffers from increased risk of miscarriage, preterm birth, and fetal distress [11], and for her being pregnant is not a protection from domestic violence [12]. A study carried out in Iran on a sample of 313 women shows that 55.9% of them experienced at least one episode of violence during their gestation. Even more relevant is the fact that 8.9% of those women were victims of violence exclusively during pregnancy [13]. Previous research has shown that violence during pregnancy is a risk factor for serious future violence, and even violent death [14]. In fact, research on women killed by their partners during pregnancy suggests that these deaths are most likely to occur early in the pregnancy, rather than later [15].

The literature on pregnancy and partner violence posits that increased stress may exacerbate intimate partner violence [16]. A number of risk factors contribute to violence during pregnancy, including: an increase in financial concerns for the male partner, doubts over the paternity of the child, decreased attentiveness (i.e., feeling sexually neglected), an increased state of affective and economic dependence, and stress due to sleepless nights [17]. It may be that the increase in sexual violence during this time is due to women being less interested than their partners in resuming sexual activity, a situation that could trigger unwanted sexual advances. Pregnancy is the trigger point for, or at least a risk factor in, domestic violence [18,19]. A possible explanation for this phenomenon is the men’s need for love exclusiveness, jealousy towards the child his partner is carrying, and at the same time the weakness and fragility of pregnant women who are less able to defend themselves. In literature, this has been attributed to the phantasmic processes activated by both parents during pregnancy.

Some risk factors often accompany the gestational period and make women particularly fragile and less able to defend themselves against violence [19]. The increase in stress leads to a decrease in coping skills and an increased level of conflict in the couple, which, in the most serious cases, results in violence; a couple relationship characterized by jealousy and suspicion can turn into a violent relationship during pregnancy, especially when the couple are also of a low socio-economic status.

Women in their first pregnancy are more likely to suffer IPV because of the higher level of stress experienced in transitioning to parenthood for the first time [20]. An increase in stress and frustration is among the reasons most frequently highlighted by the literature to explain the violent behavior of the future father towards the partner.

In Italy, ISTAT (Istituto Nazionale di Statistica) [21] data shows that 11.5% of female victims were pregnant at the time of the violence. For 50.6% of these, the frequency of violent episodes during pregnancy remained constant, for 17% it was reduced, and for 16.6% it increased. Finally, 15% say that the beginning of pregnancy coincided with the start of violent behavior by the partner. Furthermore, interviews with health personnel dealing with gender violence in the Naples area supports these international data.

Pregnancy represents a key moment for diagnosing domestic violence, and healthcare workers, therefore, ought to investigate domestic violence and provide consultation both during the pregnancy and the postnatal period [22]. Routine pregnancy screening tests are a good opportunity for health workers to take action [23,24]. Some findings [25] show that a good assessment carried out by a trained practitioner can help the victim of violence to disclose her story. Programs of intervention for pregnant women, in order to be effective, should be aimed at raising awareness on intimate partner violence, safety education, advocacy, and community referral [26].

Domestic violence is a plight that deserves further investigation if we intend to halt its spread effects as well as its prevalence [27,28]. Despite the need for employing new intervention measures and protection to limit the increase of domestic violence, a significant intervention aimed at reducing this phenomenon is yet to emerge [29]. This paper sets out an ecological and multidimensional vision of domestic violence. In doing so, it investigates the role of individual, organizational and contextual factors set up in health services that are geared in particular towards pregnant women. 

According to the ecological perspective [30,31], the individual development is the outcome of a concurrent interaction of interdependent factors (individual characteristics, causalities, and contextual features). A change in intervention procedures can help female victims of violence. This change, however, must involve both health professionals and the environment in which they work [32].

General Practitioners seem to be the first resort for abused women and victims of domestic violence and when General Practitioners recognize and actively deal with domestic violence, they represent a valid opportunity in directing women to communities and services that provide legal assistance [33,34,35]. General Practitioners, similarly to obstetricians in maternity wards limited the intervention to the mere physical aspects of the case and in doing so they protected themselves from overwhelming emotional involvement in taking in stories of domestic violence [34]. A further aspect that emerged from the interviews is a lack of knowledge about the way other services tackle the issue of domestic violence. Given the lack of shared procedures and protocols among health and social workers, domestic violence management is based only on personal friendship, support, and solidarity among professionals [36]. The fragmented nature of domestic violence management is due to a lack of coordination and integration between diverse models of conceptualization of violence itself, lack of connection among stakeholders, and lack of intervention strategies [37]. The health workers’ emotional hurdle, along with the fragmentation of their interventions, represents a weakness as well as a risk factor, in that they force women to deal with the issue of violence on their own as if this were a wholly private matter. Research shows that health workers tend to approach domestic violence with feelings of inadequacy and frustration, due to a lack of training or awareness over the causes and effects of this phenomenon [38,39,40,41,42,43,44,45,46,47,48,49,50].

Health and social workers themselves sometimes hold many of the most common prejudices related to women who have experienced violence, and may even regard violence as the consequence of a provocative attitude on the part of the women [51,52,53]. On these grounds, the social workers prefer not to undertake the path of intervention [54,55], and they most commonly use only a supportive approach, in particular when the partner is the perpetrator [56]. Despite the difficulties experienced by health operators in relating to women victims of intimate partner violence, it is important for them to learn how to recognize and handle cases of violence, since women ask formal services such as police, doctors and nurses for help only when the violence becomes more severe [57].

As several studies have highlighted [58,59,60], the emotional hurdle experienced by health workers is usually followed by a woman’s reluctance to disclose violence to them. In turn, health workers do not ask those specific questions that would help victims open up.

Women victims of violence were eager to talk about their experiences [61], but health workers are not always able to deal with this issue without further victimizing the woman. In some cases, in a collusive framework, health professionals think that women do not want to speak about violence, so they prefer not to ask straightforward questions regarding their experiences [44,62,63,64]. As a result, it is necessary for health professionals to learn how to use the control approach. This model, in fact, emphasizes the importance of taking into account the responsibility of the aggressor, and uses legal means of intervention [65], proposing special procedures in the emergency department to support women in reporting psychological and physical abuse [66]. Many women come to the ED (Emergency Department) with severe injuries, but the link between injuries and domestic violence is often not recognised and women may not receive appropriate treatment [67]. Therefore, it is necessary to equip health professionals with the proper skills to increase their awareness and understanding of the dynamics of domestic violence, and to develop procedures for handling such cases in the most effective way.

The literature [68] shows, moreover, that women prefer more informal interventions and individual counseling. In particular, they seem to prefer interventions that: are not followed by a divulgation of information; do not label the person as a victim of IPV; provide a larger range of options; and that are respectful of self-autonomy. Furthermore, the literature reports that victims of violence that had positive encounters with health and social workers had a series of benefits in terms of their future well-being [60,69,70].

As far as the health care routine and the relationship among the medical personnel is concerned [71], women believe that, in cases where they experienced abuse, it is important that personnel ask them straightforward questions, just as they do when they investigate any other case history. Among the individual factors that help patients talk about their experience of gender-based violence, we stress asking straightforwardly, being truly interested in getting to know about the abuse, and not treating the patient superficially or lightly. This approach helps them reveal the violence without treating it is as a taboo.

In the light of the literature, our study investigates the way professionals such as nurses, gynecologists, and midwives approach intimate partner violence. The Obstetric-Gynecologic Walk-in Clinic is a service that a pregnant woman can refer to when she has concerns over her own health and/or that of her child. In cases of domestic violence, a gynecological examination represents a good opportunity for health workers to uncover the issue [72].

This study highlights the hurdle of grappling with domestic violence within a specific hospital ward, an Obstetrical-Gynecological walk-in clinic in the south of Italy. Here, professionals during their working practice usually deal with and have access to both the women’s own private lives and that of the couple. Therefore, this study intends to give a specific contribution to the literature by (a) examining the representations and meanings that health personnel hold about domestic violence; (b) investigating the professional experience that they have gained from working with women in general, and those who have been victims of domestic violence in particular, and (c) adding an understanding of what kind of resources measures the health personnel working in this specific setting can introduce in order to contrast and reduce the impact of domestic violence.

In order to increase the level of confidence in the results and to resolve methodological biases [73,74], as well as to determine whether the phenomenon under examination has been accurately framed, we adopted two kinds of analysis, that is: a symbolic-structural semiotic analysis, and a qualitative content analysis.

## 2. Materials and Methods

### 2.1. Participants

The health personnel (24 females and 6 males) of an Obstetrical-Gynecological walk-in clinic in the south of Italy were invited to become involved in this study. The entire staff of the ward, which includes 10 obstetricians and nurses, 6 trainees, 6 medical students, 1 health practitioner, 4 gynecologists, and 3 first aid police officers, were interviewed. The average age for the interviewees was 36.5 (SD = 14.09). The number of years of service ranged from 1 to 37, with a mean of 12.9 (SD = 13.66). Moreover, the interviewees have been working at this particular Obstetrical-Gynecological walk-in clinic for differing lengths of time: from a minimum of 1 month to a maximum of 37 years (mean 10.03; SD = 11.98). 

In order to carry out the following semeiotic analysis, we used the above-mentioned demographic variables. These were used as a means for interpreting and analyzing possible interdependent ties. For this analysis, gender was categorized as male or female. Age was also categorized into two groups, 21–38 and 50–61. The length of service was broken down into four ranges, that is: 1–9 years, 10–20 years, 21–30 years, and 31–40 years. The length of service at this particular clinic was similarly divided into four categories: 1–7 months, 1–20 years, 21–30 years, and 31–40 years.

### 2.2. Procedures

Thanks to the collaboration and authorization provided by the Director of the Gynecological Unit of the Azienda Ospedaliera Universitaria (AOU) Federico II, we administered ad-hoc interviews aimed at exploring some complex dynamics that would not otherwise be directly observable; the research team prepared an interview grid that focalized the themes to explore, but not fixed questions.

Each interviewee was contacted in advance by phone in order to arrange an appointment consistent with his/her time and working schedule. Subsequently, they were asked to sign their informed consent for the study. The working schedule of the professionals turned out to be an obstacle whereby the textual data collection was delayed over a period of time longer than originally expected. During this time (from January to April), we tried to balance the “participant observation” and the “observing participation” [74]. Four young researchers conducted interviews with users of the walk-in clinic and patients at social risk and performed ethnographic observations in the emergency treatment waiting room. These reports were useful to the research team to better understand and code the health workers’ interview transcripts.

### 2.3. Measures

While the research used a narrative approach, the interviews were focused on specific topics consistent with guidelines for focused interview [74].

This is an approach that gives the interviewee the opportunity to express his/her experience and visions on the research issue, but at the same time allows the researcher to explore specific areas to take under consideration.

Here the interviewers gathered data that fell into three thematic areas (a) the representation and meaning that personnel hold about domestic violence, especially during pregnancy; (b) the professional experience they have developed in dealing with women’s issues, specifically with victims of domestic violence; and (c) the expectations that the personnel employed in the Obstetrical-Gynecological walk-in clinic have on the medical care in regards to the way the service they work for can tackle domestic violence against women.

### 2.4. Data Analysis

The textual data has been complementarily analyzed by means of two different procedures: (a) Symbolic-structural semiotic analysis and (b) Thematic content analysis.

(a) Symbolic-structural semiotic analysis. The Symbolic-structural semiotic analysis investigated the underlying framework of the discourse [75,76]. This allowed us to obtain a synthetic representation of the textual content while defining the meaning that our interviewees attribute to domestic violence.

Semiotic analysis is a textual approach that attempts to break up, as well as highlight, the meaningful framework of a text [77]. By means of this process, semiotic analysis reveals the deep rules and meaning underpinning a text while comparing them with other texts. Meanwhile it allows for the cultural inter-textual setting in which the texts are embedded by taking into account the particular historical moment and society in which they occur. According to the structuralist approach, in fact, the meanings and norms that rule a text cannot be studied in isolation. They are always part of a context of interconnected semantic relations [78].

In order to carry out our analysis, T-LAB software was employed [79]. The analysis of elementary contexts that T-LAB carries out as Cluster Analysis is a multivariate analysis. This allowed us to explore as well as to show the way the content of the corpus is represented by the thematic clusters. Each, in fact, is a) composed of a set of elementary contexts, which are characterized by the same patterns of key-words; and (b) described through lexical units and those variables that better describe the constituting elementary contexts. In some way, we could state that this kind of analysis provides a map of isotopies (iso = the same; topos = place), that is, either “general” or “specific” themes that are characterized by the co-occurrence of certain semantic traits [79]. The clusters have been labeled according to the words that constitute them. Some illustrative variables (i.e., sex, age, profession, and years of service at the clinic) have also been included in the textual analysis in order to verify to which cluster these are more frequently associated.

The interviews were fully transcribed and the textual material was analyzed by means of a structural-symbolic quali-quantitative methodology [80]. T-LAB provided a set of tools for textual analysis by “extracting, comparing and graphically depicting” [79] the content of the interviews. The latter were transformed into a single corpus, which was segmented along socio-demographic variables, and then treated with the process of disambiguation in order to resolve word sense ambiguity cases. We then turned to the selection of key-words and the lemmatization (by, for example, taking back verbs to their infinitive form, adjectives and nouns to the masculine singular form, prepositions to their base form, and so on). In doing so we managed to generate the so-called “lemmas”. These have contributed to building a lexical dictionary. This dictionary constitutes the basis upon which we have carried out our analysis.

Our aim was to obtain a synthetic representation of the textual content while performing co-occurrence analysis of key words to identify thematic clusters of context units.

(b) Thematic content analysis. For the Qualitative Content analysis, the interviews were broken up with the aim of obtaining the simplest constituents using a set of pre-defined categories [81,82]. This is aimed at building intuitive inferences over the meaning of the interviews rather than measuring the frequency of different categories.

These categories were pre-defined according to the purposes of this study, that is investigating women’s and health personnel’s individual features, the institutional and networking organization of the service concerned with fighting domestic violence. We also used a deductive analysis [83] to integrate theory driven code—based on the research question and the theoretical framework—with data-driven ones [84]. 

Categories constitute the bedrock of the analysis and because in our case the chosen extracts are illustrative of the analytic points the researcher makes about the data, they illustrate or support an analysis “that goes beyond their specific content, to make sense of the data” [85], (p.25). That is precisely why, consistent with the latter, we would specifically refer to thematic content analysis.

As the encoders were usually the most affecting variable the interviews were codified by three independent judges and the transcripts were further categorized in a wider team discussion.

## 3. Results

### 3.1. Symbolic-Structural Semiotic Analysis

The corpus we explored is made up of 19,739 occurrences, of which 2182 are distinctive words. Within the corpus, 423 clusters of context units were identified (see Table 1).

Co-occurrence Analysis. By analyzing the kind of language utilized by the interviewees, the co-occurrence analysis focused on co-occurrences and similarities that determine the contextual meaning of the key-words selected. Therefore, the following word-associations were examined: I, we, woman, violence, and psychologist (see Table 2). The lemmas analyzed were selected in accordance to the purposes of the present study. 

The lemma “I” is one of the most frequently recurring lemmas in the corpus. It is mentioned in association with words like see, think, patient, being aware, speak, being subjected, personally and woman. The presence of words like think, know, being subjected to, and think, conjure up the image of a “wary health personnel”, which is enmeshed in thinking over, knowing, and pondering. It is worth noticing the presence of the phrase “being subjected to” along with the absence of other action-oriented verbs. This highlights that the personnel feel not only wary, but also powerless. 

The lemma “we” co-occurs with the word obstetrician, student, trainee, manage, speak, find, procedure, and ours. The first three words refer to some of the key figures who work in the clinic. It is interesting to notice the presence of verbs such as manage, speak, and find, which stand in contrast with the personal pronoun “I” in that they refer to a shared working practice. 

The lemma “woman” conveys the kind of image that health personnel in the clinic hold of the female sex. The co-occurrence of words like hide, tolerate, husband, children, accept, see, happen, and violence depicts the image of a woman, both as wife and mother, who bears, hides and tolerates the inflicted violence.

The lemma “violence” co-occurs with I, woman, man, child, husband, being subjected to, tell, baby, and imagine. This portrays a picture of domestic violence and its key characters: I, child, and husband. It is worth noticing that the verbs more frequently linked to the word violence here are: tell, being subjected to, and represent. What is striking here is the recurrence of the lemma being subjected to, as if violence is something that a person can only be subject to and not escape.

The lemma “psychologist” co-occurs with gynecologist, obstetrician, intervene, report, contact, approach, search, and patient. This suggests that the interviewees hold an image of the psychologist as an important and sympathetic professional figure. Patients endorsed its relevance in that the support provided by the psychologist seems to spur them to press changes. It also seems to be depicted as a supportive figure endowed with “kindness”.

### 3.2. Thematic Analysis of Elementary Contexts

For this analysis the corpus has been processed in order to pool together a number of words related to each cluster. Four clusters have emerged as a result of this operation. Each of them is characterized by a set of co-occurring lemmas, in other words, lemmas that have been used in the interviewees’ narratives.

The clusters are ordered here by threshold value, from highest to lowest, rather than cluster number.

#### 3.2.1. The Torture of Choice (CL3–158 Elementary Contexts)

Obstetricians are placed in the third cluster, with a relevance value (score) of 97.733, aged between 50 and 61 years, with professional experience at the clinic ranging from 21 to 30 years and an overall working experience ranging from 31 and 40 years. This makes up the lemmas of the third cluster (violence, surely, think, woman, search, child, press charge, her, question, happen, imagine, I, tolerate, want, person, fear, suspect, normal, being subjected to violence, know). They refer to a torn woman enmeshed in a critical state of affairs. This kind of torment is suggested by the coexistence of words like press charges and tolerate. The analysis of cluster units shows how much health personnel try to be prompt and sympathetic whenever a victim of violence steps into the clinic and needs someone to talk to:

“*We are talking about a woman who is going through incredible pain and is living a hell of a life. It must be really hard for her to press charges. We could speak to the patient to see if her husband needs psychotherapy and perhaps, he could be treated*” (F, trainee, first year of experience at the clinic).

#### 3.2.2. Organization of the Context (CL4–98 Elementary Contexts)

The fourth cluster is that of the trainees, with a score of 50.445, aged between 21 and 38 years, with less than one year of professional experience at the clinic and an overall working experience ranging between 1 and 10 years. This is the cluster that gathers the following lemmas: structure, walk-in clinic, better, sample taking, lack, delivery room, settle, operate, send, characteristic, work, personnel, room, control, gynecologic walk-in clinic, approach, childbirth, vagina, devote, trainee. These refer to the structural organization of the clinic. The analyses of the context units show the desire of the health personnel to achieve better structural planning. It is worth noticing that the word “violence” is absent here. This seems to suggest that this could be considered a contingent topic in terms of a possible re-planning for the health organization:

“*This is the only walk-in clinic that takes on every patient that steps in. That does not happen in the other walk-in centers. They could improve the looks of this place as well as its organization.*” (F, physician, third year of experience at the clinic).

#### 3.2.3. All is Tolerable in a Relationship (CL2–89 Elementary Contexts)

This is the label of the second cluster. Obstetricians are placed here, with a score of 62.225, aged between 21 and 38, with less than one year of professional experience at the clinic and an overall working experience ranging between 1 and 10 years. It includes the following lemmas you, time, partner, relationship, situation, different, depend, respect, guilt, a bit, environment, recognize, create, justify, tranquil, particular, happen, leave, case, recover, pleasure. These lemmas frame a picture of the woman in her relationship with both her partner and the health personnel. Both the analysis of the lemmas and the context units show the image of a tolerant woman who, in dealing with the abusive relationship, tends to justify the aggressor and blame herself. In the relationship with the health personnel, instead, it emerges that she can feel more at ease. However, it would be necessary to work in a quieter place, and that is exactly what the clinic lacks:

“*I think that a woman always tends to justify her aggressor, especially when it comes to her partner […] unless that involves a stranger, someone she does not know.*” (F, undergraduate student, first year of experience at the clinic).

#### 3.2.4. A Place for Urgency (CL1–81 Elementary Contexts)

This is the label chosen to describe the first cluster. This cluster is comprised of interviewees in ongoing training, aged between 21 and 38 years, with less than one year of professional experience (score of 107.739). It includes lemmas such as: obstetrician, find, we, prepare, die, alive, welcome, start, evaluate, urgency, walk-in clinic, gynecologist, go through, concern, figure, procedure, prepared, doctor, first. These lemmas refer to the image that health personnel have of their working place.

The analysis of context units shows that the lemma urgency is linked to the condition of being faced with matters of life or death. These are, indeed, the kind of issues that the health personnel have to deal with on a daily basis, due to a lack of some essential procedures (i.e., lack of collaboration, absence of violence kits). In cases of emergency, two professional figures in particular seem to be regarded as significant, that is the gynecologist and the obstetrician. The gynecologist is the first person to be summoned when a patient enters the walk-in clinic. This professional has the task of taking on, making an initial assessment of, and then directing the patient to the most suitable ward (i.e., delivery room).

Obstetricians are, instead, more underestimated. They are called upon only in the absence of the gynecologist. The walk-in clinic is a place of transit, in that patients come and go. In case a patient who has been subjected to abuse comes along it would be necessary to deliver some first aid, or to administer a vaginal swab test.

“*The thing is that we don’t have any, we’ve got only the rectal ones, which we administer to all pregnant women*” (F, undergraduate student, first year of experience in the clinic).

### 3.3. Thematic Content Analysis

In order to analyze the health personnel’s concerns for those resources that in their view, are able to reduce the risk of domestic violence while developing interventions within the health service, we further analyzed the content of selected categories as mentioned: women’s individual features, health personnel’s individual features, and the institutional and networking organization of the service. However, these factors, which emerged with regard to the organization of the health service, are described as ‘needed’, in that the health service does not provide them yet.

#### 3.3.1. Women’ Individual Resources: Self-Respect, Self-Esteem and Autonomy 

The personnel identify self-respect, self-esteem, and autonomy as the necessary resources for a woman to disentangle herself from the violent relationship: 

“*I think they struggle to stand up because they lack self-esteem. It would be necessary to increase their level of self-esteem and, if very poor, try and make them independent with a job*” (M, 28 years, Physician, 4 years of service at the clinic).

“*Of course, she’s got to respect herself and expect respect in turn, and she’s got to be aware that she deserves respect*” (F, 27 years old, Trainee, 1 year of service at the clinic).

#### 3.3.2. Personnel’s Individual Resources

The personnel offer all their availability and empathic listening as a resource for providing a protective environment for the victim. On one hand, this resource predisposes a welcoming and intimate setting. On the other hand, however, this is not followed by guidelines and support.

“*I was there for this lady. We talked a little and then changed the subject. If I have a hunch that she does not want to talk it out, I am not going to pry. You always try to be there though, and let her talk. If she doesn’t feel like talking, I am still there for her*” (F, 23 years, undergraduate student, 3 years of service at the clinic).

“*[…] it depends on what kind of person she is, how she talks to you, how she tries to make you understand. I don’t know really. I think I’d try to make her say the right thing, because they hide it many times. I’d try to let her speak, to give her some advice, that’s what I’d do*” (F, 58 years old, Obstetrician, 37 years of service at the clinic).

#### 3.3.3. Resources at the Organizational Level

(a) trained practitioner

The health personnel of the clinic regard the figure of the trained practitioner as a protective factor whereby the service can manage the issue of domestic violence. The health personnel believe that training on the management of domestic violence is paramount if they are to become more aware of how to intervene as well as more able to deal with their emotional resistance arising from the very nature of domestic violence:

“*First of all, we need the health personnel (physicians, obstetricians, nurses) to be well-trained. And then everything else... tools like the rape kit. I mean, I don’t even know these things, so first and foremost information and training and then availability of tools*” (F, 27 years, Trainee, 2 years of experience at the clinic).

“*We need from our institutions (the school of obstetrics, the course of obstetrics, the specialization school, the specialization program that our trainees do, the directors of the programs) a higher interest in this issue. Perhaps that could give us even more suitable tools, I don’t know, like training, courses, whatever. I think this is what we need*” (F, 21 years old, undergraduate student, 3 years of experience at the clinic).

This is a good example of what it means when they state that the clinicians are aware of the need to increase their knowledge in relation to the violence [86].

(b) sympathetic service 

The possibility of providing a welcoming room to take in the women is regarded as a further protective factor for the clinic. To have use of a room for the interviews means to feel comfortable in a confidential space where time can be devoted to the patient away from the hustle and bustle of the emergency room. This could allow the personnel to give a thorough assessment of the possible indicators of violence while actively engaging in a conversation with the patient in order to delve into her story:

“*I need like a quieter place, like my own room, my own place, my own environment that would not be yet another cold place, though still in a clinical setting. But I need a room that could be a bit more, I don’t know, colorful, lively basically. So, when a patient steps in she can find a place a bit more positive, welcoming, quiet, like it could give her a bit of peace of mind*” (F, 50 years old, Obstetrician, 20 years of service at the clinic).

*“We need the “rape kit” and a more private and confidential room”* (F, 35 years old, Obstetrician, 10 years of service at the clinic).

(c) health service as an informative place 

It is necessary to set up visible and usable information desks geared towards women and providing useful information such as leaflets, and strategically located pictures:

“*Here we are, always up and running and available. It’s just we don’t have the tools, we don’t get the opportunity to provide a perfect service* (F, 27 years old, trainee gynecologist, 2 years of service at the clinic).

“*It is unlikely that a woman would come in today and tell you […] they tend to hide it […] it would be good to have a waiting room with informative posters and/or telephone numbers of specific services to which victims of violence can turn to*” (F, 30 years old, Obstetrician, 3 months of service at the clinic).

(d) violence detection procedures and the “rape kit”

Being able to use a series of tools for detecting the signs of a woman subjected to physical or sexual violence constitutes a valuable, responsive and protective factor for the victims. The use of a diagnostic “rape kit” requires adequate training and a clear as well as shared understanding among the personnel of how detection procedures work. Moreover, the possibility of using adequate diagnostic equipment allows the victim to be treated with straightforward protective interventions and avoid the risk of being directed to other local services, which would further increase her disorientation:

“*[…] to have a kit that allows us to detect some proof, otherwise we are only going to make things worse. Through the examination with the speculum, we can extract some sample sperm*” (F, 31 years old, Trainee, 5 years of service at the clinic).

“*We are not ready at all. We should perform all the vaginal sampling, both for infection and DNA sampling, then the sample storage […] of course we are not in the position to do all these things, I mean, I don’t think we have even the suitable equipment*” (F, 28 years old, gynecologist trainee, 2 years of service at the clinic); and

(e) psychologists to take care of the victim

In taking care of women victims of violence, the personnel regard the psychologist as a necessary professional to support the health service. The psychologist is seen as a figure capable of taking charge of the patients and understanding them: 

“*The personnel, I mean, we have physicians and obstetricians and everything but maybe a health worker like a psychologist that can listen to the patients, we aren’t in a position to do all these things, I mean, I don’t think we have patients and try to understand those who have these problems, because they rarely open up with a physician, a nurse or an obstetrician. Perhaps a psychologist here could help us out*” (F, 27 years old, Trainee, 2 years of service at the clinic).

“*We really lack the figure of the psychologists and even in the voluntary abortion service there is none. So, the gynecologist and the obstetrician have to act a bit like a psychologist too. It’s true we take some exams in psychology but those are more the basics, we can’t really make up for an actual psychologist*” (F, 60 years old, Obstetrician, 37 years of service at the clinic).

To sum up, we acknowledge that violence during pregnancy is a matter of fact; at the same time our research supports studies identifying a sort of professional ‘blindness’ and unaware silence among professionals. The dangerous effects of violence on women and child health are known, but in contact and clinical interviews with women and couples the topic is not dealt with. Therefore, the need for social awareness of the negative effects of violence should be promoted by institutional training and stronger cultural awareness. 

## 4. Discussion

The symbolic-structural semiotic analysis has revealed that the interviewees depict a victim of domestic violence that is torn by the difficulty of making the right choice, and is more inclined to forgive and justify her partner as a consequence. The personnel believe that in cases of violence, a woman can draw on her self-esteem and, above all, the ability to respect herself and, in turn, ask for respect. This shows that our interviewees place the safeguard of the couple’s relationship not only on love, but also on the right to mutual respect.

The analysis of the data reveals that the interviewees of the clinic do not regard the issue of domestic violence as a matter of direct interest for the health service. The clinic is seen as a place for urgent contact, but one where there is not enough time to dedicate to this kind of patient, nor an adequate space to care for and listen to them. Our results appear to be in line with literature. The scientific research reports limited use of violence screening in maternity wings [38], as well as a negative attitude by obstetricians when it comes to including questions regarding violence and abuse in pre-natal reports [39].

Furthermore, the majority of the interviewees in our study report that they have never encountered a full-blown case of a victim of violence. At the same time, it seems that the possibility of coming across signals that could raise suspicion of episodes of violence against women constitutes a reason for concern by the personnel of the clinic. When it comes to facing cases of violence, the personnel are not equipped with a straightforward screening procedure and protocol to detect domestic violence (let alone diagnostic rape kits—which they consider useful tools that the services should provide). In fact, they resort to personal listening skills: and an open attitude towards the other. Individual resources seem to be the only factors available in this health service.

It is interesting here to refer again to the word associations given by the lemma “I”, which show the personnel who, despite relational and listening skills, feel helpless and unable to offer help to the victim. The lack of policies and organizational protocols among the services [87] to address domestic violence also influence workers’ beliefs about their intervention towards victims of domestic violence [88].

The personnel regard training, the availability of suitable interview rooms, and diagnostic procedures and tools as protective factors, which should be provided by the clinic. When these resources are lacking, the personnel feel helpless in dealing with the victim of violence in that they can only offer to listen to her. This alone is not enough in supporting the woman to report the violence or to help her plan a new life and, in doing so, disentangle herself from the violent state of affairs [67]. Therefore, the Obstetrical-Gynecological walk-in clinic we have examined is not an effective resource for victims of violence. Results show that this specific context is still far from taking a straightforward stance with regard to the management and prevention of domestic violence. On the contrary, the context itself proves to be a vulnerable risk factor for domestic violence. The respondents of our research showed a vulnerability of the Obstetrical-Gynecological ward as well as its potential for combating violence during pregnancy. Domestic violence should be a specific topic in basic training of health personnel. Especially in the 90s [10,15,44] there has been research on the dangerous effects of violence during pregnancy on the child and mother’s health. Furthermore, research on domestic violence should better focus on the psychic dimension of violence during pregnancy and its effect on the mother/child bond and on the couple’s future relationship.

Authors should discuss the results and how they can be interpreted in perspective of previous studies and of the working hypotheses. The findings and their implications should be discussed in the broadest context possible. Future research directions may also be highlighted.

## 5. Conclusions

The literature [89,90] has highlighted described the poor diffusion of good practices and strategies of individual and organizational empowerment employed by health and social workers in dealing with domestic violence. However, there seems to be an increasing interest in taking charge of victims of violence in social and health institutions. Consistent with recent research [32,91] as well as with what has been reported by our interviewees, the key components to help women victims of violence are: (a) empowering health providers to take initiative to address IPV in the health care setting; (b) recruiting selected staff that have received in-depth training on IPV; (c) providing training to every member of the clinical staff interacting with patients (doctors, nurses and administrative staff); (d) periodic monitoring to improve the system’s response; e) developing a new clinical culture, with the inclusion of new professionals, which supports continuous improvement. Even if in their words there is a lack of a straightforward willingness to become directly involved with the police force and women’s shelters, the results of our study have raised awareness over the necessity of rape kits, training of personnel, and effective intervention strategies and as well as procedures introducing also psychologists.

Thanks to a recent regional law in the Campania Region to act against domestic violence, these suggestions could be brought to bear. Even if this study only takes into account the gynecological walk-in clinic within a larger university hospital, the results it provides can be a useful guideline for larger scale interventions in Obstetrical and Gynecological centers as well as for the training of the personnel appointed for the care of pregnant women.

## Figures and Tables

**Table 1 ijerph-16-03087-t001:** shows the specific features of the corpus.

Key Features of the Corpus	Number
Texts	30
Contexts	426
Occurrences	19,739
Lemmas	2008
Words	2182
Frequency Threshold	4
Key-words	302

**Table 2 ijerph-16-03087-t002:** Chi-square values (*χ*^2^) for the co-occurrences of the lemmas I, We, woman, violence, and psychologist.

Specific Lemmas	Co-Occurrences	*χ* ^2^
I	See	16.41
	Know	4.08
	Patient	1.07
	Think	2.99
	Personally	14.13
	Woman	0.95
	Learn	10.26
	Talk	0.41
	Violence	2.80
	Being subjected	2.89
We	Obstetrician	5.62
	Manage	10.60
	Talk	0.746
	Our	4.32
	Find	2.99
	Proceed	11.23
	Student	9.06
	Trainee	6.81
Woman	Husband	35.81
	See	4.10
	Being subjected	12.89
	Hide	15.07
	Happen	0.97
	Accept	11.87
	Child	6.13
Violence	Being subjected	21.70
	Child	17.99
	Man	18.64
	Woman	6.56
	Child	12.39
	I	2.80
	Tell	6.56
	Husband	5.83
Psychologist	Approach	14.09
	Contact	14.09
	Report	9.43
	Gynecologist	9.07
	Intervene	9.07
	Search	4.63
	Patient	0.43
	Obstetrician	3.39

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
