# Peer review of "Healthcare Professionals’ Perceptions and Concerns towards Domestic Violence during Pregnancy in Southern Italy"

_ijerph, 2019, doi:10.3390/ijerph16173087_

Round 1

Reviewer 1 Report

The article is very well written from the abstract to the conclusions.

It is extremely important that strategies for discovery of intimate partner violence are stimulated

The biggest problem encountered was with references: 27 references were cited from 1979 to 1999, 40 references from 2000 to 2009, and 24 references from 2010 to 2019. Although we know that there are references that even old ones are fundamental, I believe it would be interesting the authors to cite more recent references. I suggest updating the literature review.

Author Response

Thanks, we updated the literature.

Reviewer 2 Report

Abstract:

-“A prospective study of pregnancy highlighted a large number of women abused by their partners, especially among low-income teenagers attending hospital for pregnancy check-ups.” Are you referring to the literature or introducing the current study? The positioning of that sentence leaves one to wonder…

-The interviewees, from an Obstetrical- Gynecological walk-in Clinic in the south of Italy. As this stands, this is a sentence fragment. Add the word “were” in front of “from”.

--How many professionals were interviewed?

--Research can’t “report” but people can…

Introduction:

--“There is in fact an increase evidence for the violence perpetrated against women during this period.” Where is the reference for this?

--“While only a few studies have followed pregnant women prospectively to examine the impact of violence on birth outcome.” This sentence isn’t complete.

---“and being pregnant provided no protection at all from domestic violence” This is what makes this research important and interesting.

---“Pregnancy is the trigger point for,” Rather, “Pregnancy is a trigger point for..”

---“A possible explanation for this phenomenon…In literature, this has been attributed..” Do you have references for these statements? There are several similar instances where assertions are not backed up with references….

--“if we intend to halt its spread” This phrasing makes it seem like an infectious disease…

--“General Practitioners limited the intervention to the mere physical aspects of the case” This is similar to the postpartum when the clinical outcomes are the focus but the mothers mental health is not

---“(c) adding an understanding of what kind of resources the health personnel working in this specific setting introduce in order to contrast and reduce the impact of domestic violence.” Can you clarify…this part of the study aim is hard to understand…

Methods:

--Respectable sample size for a qualitative study and good that you interviewed all professionals who would have contact with the women

---“we administered ad-hoc interviews” Does this mean that there were no prescribed interview questions?

---similar to earlier on, this part of the study aims is wordy and difficult to digest: “the expectations that the personnel employed in the Obstetrical-Gynecological walk-in clinic have with regard to the way the service they work for can tackle domestic violence against women.”

--For the themes underneath “Thematic Content Analysis,” (ie, “Womens’ Individual Resources”) employ spaces in between themes so that the themes stand out to the reader. As it stands, all the themes are jammed together.

---The whole manuscript should be checked for grammar, word choice, etc eg, “This is a good example of what mean” There is a word missing here.

--This idea of having an on-site psychologist is true for maternal mental health issues like depression and anxiety also – always more effective to have on-site mental health counseling available.

Discussion:

--Can you break this sentence up into two sentences? “When it comes to facing cases of violence, as the

personnel are not equipped with a straightforward screening procedure and protocol to detect

domestic violence, let alone diagnostic rape kits – which they consider useful tools that the services

should provide – they resort to personal listening skills and an open attitude towards the other.”

--“This alone is not enough in supporting the woman to report the violence or to help her plan a new life and, in doing so, disentangle herself from the violent state of affairs.” Do you have a reference for this?

--“Authors should discuss the results and how they can be interpreted in perspective of previous

studies and of the working hypotheses. The findings and their implications should be discussed in

the broadest context possible. Future research directions may also be highlighted.” Is this part of the narrative ? Looks like instructions?

---Literature can’t “highlight” but researchers can – avoid personification of inanimate objects.

Author Response

Abstract:

-“A prospective study of pregnancy highlighted a large number of women abused by their partners, especially among low-income teenagers attending hospital for pregnancy check-ups.” Are you referring to the literature or introducing the current study? The positioning of that sentence leaves one to wonder…

We clarified .it

-The interviewees, from an Obstetrical- Gynecological walk-in Clinic in the south of Italy. As this stands, this is a sentence fragment. Add the word “were” in front of “from”.

We added it

-How many professionals were interviewed?

We added it

-Research can’t “report” but people can…

We changed, as suggested

Introduction:

--“There is in fact an increase evidence for the violence perpetrated against women during this period.” Where is the reference for this?

We moved it from the next paragraph

-“While only a few studies have followed pregnant women prospectively to examine the impact of violence on birth outcome.” This sentence isn’t complete.

We completed it

-“and being pregnant provided no protection at all from domestic violence” This is what makes this research important and interesting.

We agree with your suggestion

-“Pregnancy is the trigger point for,” Rather, “Pregnancy is a trigger point for..”

We modified it, as suggested

“A possible explanation for this phenomenon…In literature, this has been attributed..” Do you have references for these statements? There are several similar instances where assertions are not backed up with references….

We improved the text

-“if we intend to halt its spread” This phrasing makes it seem like an infectious disease

We changedin ‘effects’

“General Practitioners limited the intervention to the mere physical aspects of the case” This is similar to the postpartum when the clinical outcomes are the focus but the mothers mental health is not

Yes, we agree, but there are no references in that sense

“(c) adding an understanding of what kind of resources the health personnel working in this specific setting introduce in order to contrast and reduce the impact of domestic violence.” Can you clarify…this part of the study aim is hard to understand…

Please, see now

Methods:

-Respectable sample size for a qualitative study and good that you interviewed all professionals who would have contact with the women

Yes, thanks

“we administered ad-hoc interviews” Does this mean that there were no prescribed interview questions?

 Yes there was an interview grid focalizing the themes to explore, but not fixed questions

- similar to earlier on, this part of the study aims is wordy and difficult to digest: “the expectations that the personnel employed in the Obstetrical-Gynecological walk-in clinic have with regard to the way the service they work for can tackle domestic violence against women.”

 Ok we improved

For the themes underneath “Thematic Content Analysis,” (ie, “Womens’ Individual Resources”) employ spaces in between themes so that the themes stand out to the reader. As it stands, all the themes are jammed together.

Ok we did it in all 3.3

The whole manuscript should be checked for grammar, word choice, etc eg, “This is a good example of what mean” There is a word missing here.

Ok we did itin all the text

This idea of having an on-site psychologist is true for maternal mental health issues like depression and anxiety also – always more effective to have on-site mental health counseling available.

We emphasized it

Discussion:

Can you break this sentence up into two sentences? “When it comes to facing cases of violence, as thepersonnel are not equipped with a straightforward screening procedure and protocol to detect domestic violence, let alone diagnostic rape kits – which they consider useful tools that the services should provide – they resort to personal listening skills and an open attitude towards the other.”

Yes we did it.

“This alone is not enough in supporting the woman to report the violence or to help her plan a new life and, in doing so, disentangle herself from the violent state of affairs.” Do you have a reference for this?

Yes we introduced it

-“Authors should discuss the results and how they can be interpreted in perspective of previous studies and of the working hypotheses. The findings and their implications should be discussed in the broadest context possible. Future research directions may also be highlighted.” Is this part of the narrative? Looks like instructions?

Please, see now

-Literature can’t “highlight” but researchers can – avoid personification of inanimate objects.

We improved the text